# Iron-Chelation Treatment by Novel Thiosemicarbazone Targets Major Signaling Pathways in Neuroblastoma

**DOI:** 10.3390/ijms23010376

**Published:** 2021-12-29

**Authors:** Peter Macsek, Jan Skoda, Maria Krchniakova, Jakub Neradil, Renata Veselska

**Affiliations:** 1Laboratory of Tumor Biology, Department of Experimental Biology, Faculty of Science, Masaryk University, 601 77 Brno, Czech Republic; macsek@mail.muni.cz (P.M.); janskoda@sci.muni.cz (J.S.); maria.krchniakova@mail.muni.cz (M.K.); veselska@sci.muni.cz (R.V.); 2International Clinical Research Center, St. Anne’s University Hospital, 656 91 Brno, Czech Republic; 3Department of Pediatric Oncology, Faculty of Medicine, University Hospital Brno, Masaryk University, 662 63 Brno, Czech Republic

**Keywords:** thiosemicarbazone, DpC, neuroblastoma, MYC, EGFR, NDRG1, lipid droplet

## Abstract

Despite constant advances in the field of pediatric oncology, the survival rate of high-risk neuroblastoma patients remains poor. The molecular and genetic features of neuroblastoma, such as *MYCN* amplification and stemness status, have established themselves not only as potent prognostic and predictive factors but also as intriguing targets for personalized therapy. Novel thiosemicarbazones target both total level and activity of a number of proteins involved in some of the most important signaling pathways in neuroblastoma. In this study, we found that di-2-pyridylketone 4-cyclohexyl-4-methyl-3-thiosemicarbazone (DpC) potently decreases N-MYC in *MYCN*-amplified and c-MYC in *MYCN*-nonamplified neuroblastoma cell lines. Furthermore, DpC succeeded in downregulating total EGFR and phosphorylation of its most prominent tyrosine residues through the involvement of NDRG1, a positive prognostic marker in neuroblastoma, which was markedly upregulated after thiosemicarbazone treatment. These findings could provide useful knowledge for the treatment of MYC-driven neuroblastomas that are unresponsive to conventional therapies.

## 1. Introduction

Neuroblastoma is a rare solid neuroendocrine tumor arising from neural crest derivates in the developing sympathetic nervous system. Subsequently, the adrenal medulla and sympathetic ganglia are the most common sites of the primary disease. Generally affecting infants aged 18 months or younger, neuroblastoma is traditionally categorized into three risk groups [1]. Low-risk neuroblastomas (50% of cases) often undergo spontaneous regression without the need for surgical intervention [2]. Intermediate-risk group patients, while likely to undergo surgical and chemotherapeutic intervention (usually containing carboplatin, cyclophosphamide, doxorubicin, and etoposide), still have a high 5-year overall survival (OS) rate of ~90% [1,3,4]. Treatment of high-risk patients usually involves surgical resection, chemo- and radiotherapy, differentiation therapy by isotretinoin, myeloablative therapy followed by stem cell transplantation, or additional procedures, such as immunotherapy [5]. However, the survival rate of high-risk neuroblastoma patients remains low and the 5-year OS rate has plateaued at only 50% [4].

Along with age at diagnosis, histology status, and chromosomal aberrations (1p and/or 11q deletion), *MYCN* amplification ranks as one of the most important prognostic factors in neuroblastoma [6,7,8]. Along with its structural homologs c-MYC and L-MYC, N-MYC represents one of the most potent master regulators of cell fate, proliferation, and metabolism, affecting the expression of more than 15% of all human genes [9]. Being established as one of the most prominent stem cell factors, elevated expression of MYC was linked not only to sustaining but also to inducing undifferentiated phenotype in various cell types [10]. Indeed, their reprogramming potential further signifies the oncogenic role of *MYC* genes, explaining their dysregulation in up to 70% of human cancers [11]. *MYCN* amplification is of particular interest as it accounts for approximately 20% of neuroblastoma cases and is strongly associated with a malignant course of disease and poor survival, even in localized disease. This makes N-MYC an intriguing molecular target for the treatment of *MYCN*-amplified neuroblastomas. In *MYCN*-nonamplified neuroblastomas, high c-MYC levels indicate an identical clinical outcome to that of *MYCN*-amplified, further proving their functional overlap [12] and establishing c-MYC as a hallmark of an unfavorable course of this cancer type [13]. The importance of MYC oncoproteins in cancer pathogenesis cannot be overstated as dysregulation of these transcription factors alone is sufficient to promote genome instability and initiate malignant transformation [14]. Conversely, downregulation of MYC oncoproteins leads to the induction of transient [15] or even sustained [16] loss of the neoplastic phenotype. However, given the largely “undruggable” properties of MYC, clinically applicable MYC-directed therapies remain elusive and the focus has shifted to targeting proteins regulating the stability of MYC oncoproteins, disrupting protein–protein interactions with MYC cofactors or exploiting dependencies of MYC-addicted tumor cells [17].

Thiosemicarbazone-based chelators have been the subject of study for over 15 years. These low-molecular-weight, metal-binding compounds have been shown to have promising pharmacological effects on cancer cells through a variety of mechanisms [18]. By generating oxidative stress inside lysosomes, to which they are sequestered by P-glycoprotein, novel thiosemicarbazones were reported to contribute to overcoming multidrug resistance. This is caused by the formation of redox-active complexes, which results in the induction of lysosomal membrane permeabilization [19]. On a physiological level, thiosemicarbazones were found to regulate various cancer hallmarks, such as proliferation, epithelial–mesenchymal transition, and migration, many of which are credited to thiosemicarbazones’ ability to induce N-myc downstream regulated gene 1 (NDRG1) expression [20,21,22]. NDRG1-mediated effects are particularly intriguing, since NDRG1 has proved to be a promising positive prognostic factor in neuroblastoma [23].

NDRG1 is a structural gene encoding the regulatory protein NDRG1, which belongs to the four-member NDRG family of the alpha/beta hydrolase superfamily. NDRG1, however, does not elicit hydrolytic function, and its exact mechanism of action remains largely elusive. As its name suggests, NDRG1 is suppressed by N-MYC at the transcriptional level. N-MYC binds to the NDRG1 promoter in an MIZ-1-dependent manner and effectively downregulates the expression of NDRG1 [24,25]. Importantly, NDRG1 was found to be a positive prognostic factor in neuroblastoma, regardless of the *MYCN* amplification status, which suggests a functional role of NDRG1 on its own in addition to its role as a marker of N-MYC dysregulation in neuroblastoma [23]. NDRG1 has been reported to exert both oncogenic and tumor-suppressive properties in different tumor types. NDRG1 inhibited cell proliferation and migration in endometrial carcinoma [26]; sensitized cells to TRAIL-mediated apoptosis in colorectal carcinoma [27]; and reduced EGFR and HER2 levels, dimer formation, and subsequent signalization in ovarian carcinoma [28]. In contrast, NDRG1 was linked with poor differentiation and prognosis in hepatocellular carcinoma [29] and mediated chemoresistance to alkylating agents in glioblastoma [30]. However, its correlation with better prognoses and overall patient survival in neuroblastoma makes NDRG1 a target worth investigating in this pediatric malignancy.

NDRG1 has been reported to exert broad antitumor activity by downregulating various proto-oncogenic signaling molecules, notably RAS, ERK, PI3K, AKT, and SRC [31,32,33]. Such a broad effect indicates that NDRG1 is a master regulator of a more upstream signaling protein. Consequently, the role of NDRG1 in receptor tyrosine kinase (RTK) regulation was investigated. NDRG1 was found to regulate several members of the ERBB receptor tyrosine kinase family, namely EGFR. In human pancreatic (PANC-1) and colorectal (HT-29) adenocarcinoma cell lines, NDRG1 markedly reduced total EGFR levels, inhibited EGFR dimerization and activation, and lowered EGFR phosphorylation even in the presence of EGF [34]. The tumor-suppressive functions of NDRG1 were further elucidated in pancreatic carcinoma. In the PANC-1 cell line, NDRG1 promoted EGFR downregulation by facilitating its lysosomal/endosomal degradation. This interaction was mediated by a negative EGFR regulator, MIG-6, which was in turn upregulated by increased NDRG1 levels [35].

Thiosemicarbazones have been already proved efficient against neuroblastoma cells by our group and others [22,36], which makes neuroblastoma an intriguing target for the study of the molecular effects of novel thiosemicarbazones. Our initial results in a single neuroblastoma cell line also suggest that di-2-pyridylketone 4-cyclohexyl-4-methyl-3-thiosemicarbazone (DpC) can potently induce NDRG1 expression in these cells. Here, we therefore aimed to investigate the molecular effects of DpC on a panel of neuroblastoma cell lines to evaluate its impact on some of the most prominent molecules (MYC oncoproteins, NDRG1, and EGFR) in neuroblastoma.

## 2. Results

### 2.1. DpC Inhibits EGFR and Its Phosphorylated Forms through NDRG1

Initially, we evaluated the dose-dependent potency of DpC to induce cell death in various neuroblastoma cell lines and found that 20 μM DpC induces a similar response in both the *MYCN*-nonamplified cell line SH-SY5Y and the *MYCN*-amplified cell line SK-N-BE(2), killing half of the cells within 24 h (Appendix A). However, CHLA-15 and CHLA-20 were treated with only 2 μM DpC, as the former concentration killed nearly all the cells by the end of the first day of treatment. The 2 μM concentration was chosen as it succeeded in inducing a comparable effect on proliferation in these cell lines (Appendix A).

After establishing the potency of DpC in inducing cell death, we focused on investigating its potency in upregulating NDRG1, a positive prognostic factor in neuroblastoma. Cells were exposed to DpC in the medium for 24 and 48 h, and the changes in NDRG1 protein levels were analyzed by Western blotting. DpC efficiently induced marked NDRG1 expression across all tested cell lines (Figure 1A), as well as phosphorylated NDRG1, whose increase mimicked that of NDRG1 (Appendix A).

To determine NDRG1′s subcellular localization and validate the change in its expression in response to DpC treatment, immunofluorescence analysis was performed in SH-SY5Y neuroblastoma cells. The results were in accordance with the Western blot analysis, and highly upregulated expression of NDRG1 was observed in DpC-treated SH-SY5Y cells that exhibited both cytoplasmic and nuclear patterns of NDRG1 signal (Figure 1B).

Having confirmed the potency of DpC in inducing NDRG1, we tested its effect on EGFR signaling, as was previously reported in carcinoma models [34]. Indeed, DpC succeeded in moderately decreasing the levels of total EGFR over the course of 2 days in all tested neuroblastoma cell lines (Figure 2 and Appendix A).

Following this finding, it was crucial to determine whether the decrease in EGFR levels was mediated by NDRG1 induced after treatment with DpC. To study the role of NDRG1 in EGFR inhibition, siRNA-mediated silencing of NDRG1 expression was performed. Transfection with NDRG1-targeted siRNAs potently decreased NDRG1 protein levels in SH-SY5Y cells (Figure 3A), whereas only a moderate decrease was observed in SK-N-BE(2) cells (Figure 3B). In accordance with these observations, silencing NDRG1 in SH-SY5Y cells rescued the EGFR downregulation induced by DpC, thus confirming the role of NDRG1 in the regulation of EGFR expression (Figure 3A). However, this effect was not observed in SK-N-BE(2) cells, which showed only moderate NDRG1 silencing (Figure 3B), which suggests that the decrease in NDRG1 levels might have been insufficient to rescue EGFR expression. Alternatively, DpC could regulate EGFR through a different mechanism in this cell line. One such possibility was explored by Hossain et al. [37], elucidating mutual regulation between EGFR and N-MYC in *MYCN*-amplified neuroblastoma cell lines such as SK-N-BE(2).

Aside from the inhibitory effect of DpC on total EGFR protein levels, phosphorylated forms of the receptor were also of interest, as they confer the transduction of signals downstream of their molecular targets. Five phosphorylation sites at EGFR were selected, and the levels of their phosphorylation were determined in regard to DpC treatment in the SH-SY5Y cell line. Selected phosphotyrosine residues facilitate the transduction of signals toward some of the most prominent signaling pathways regulated by EGFR activation (Y845: SRC–FAK axis, Y992: PLC-γ–PKC axis, Y1045: c-CBL, Y1068 and Y1148: AKT-bound and RAS-bound signaling pathways) [38]. Similar to the total EGFR protein level, all examined phosphotyrosine forms were also downregulated by DpC treatment (Figure 4A). EGFR phosphorylated at residues Y845 (pY845-EGFR) and Y1148 (pY1148-EGFR) was decreased most significantly (*p* < 0.01), while the downregulation at Y1045 (pY1045-EGFR, responsible for the negative feedback loop pathway of EGFR via c-CBL) was the least prominent, albeit still significant (*p* < 0.05) (Figure 4A). Importantly, EGFR phosphorylated at Y1068 (one of the most significant phosphotyrosine residues playing a pivotal role in EGFR-mediated activation of RAS) was downregulated even when stimulated by EGF (40 ng/mL; 15 min; 37 °C). Immunofluorescence analysis showed the expected receptor internalization after its stimulation with EGF and confirmed the inhibitory effect of DpC on pY1068-EGFR in cells both unstimulated and stimulated by EGF ligand; a marked decrease in the pEGFR signal was detected, and its localization shifted from predominantly membranous to the form of a single perinuclear cluster, presumably a stress granule (Figure 4B). To quantify the effect of DpC on pEGFR (Y1068), flow cytometry analysis was performed, which confirmed a significant decrease in Y1068 phosphorylation (*p* < 0.05) (Figure 4C).

### 2.2. DpC Induces a Stress Response and the Activation of AKT Effectors

Having established the effects of DpC treatment on EGFR, a molecular target known for its broad range of effectors, proteome profiler arrays were used to identify downstream targets most affected by DpC treatment.

First, to evaluate the nature of the stress response elicited on 24 and 48 h of treatment with 20 μM DpC in the SH-SY5Y reference cell line, the Proteome Profiler Human Cell Stress Kit was used (Figure 5A). As expected, a sharp increase in HIF1A was detected due to DpC’s iron-chelation activity, reaffirming its role in NDRG1 activation, as well as the activation of other hypoxia-regulated proteins, such as carbonic anhydrase. This was further confirmed by immunofluorescence assay, reporting a sharp increase in HIF1A nuclear localization (Appendix A), supporting the previous studies reporting that HIF1 drives the expression of NDRG1 [39,40]. Furthermore, the response exhibited a distinct temporal variation (Figure 5B): increases in p27, SOD2, NFκB, and SIRT2 were characteristic of the early phases of the cell stress response (Figure 5A, 24 h), the latter three being particularly attributed to the oxidative stress response [41,42]. Gradual upregulation of HSP60, HIF2A, and Cited-2 as well as phosphorylated HSP27 (S78/S82), p38 (T180/Y182), and p53 (S46) defined the later stages of the stress response to 48 h DpC treatment (Figure 5A, 48 h).

Second, to examine the differential phosphorylation rate of proteins that might be regulated downstream of RTKs (including EGFR) in response to DpC treatment, the Proteome Profiler Human Phospho-Kinase Array Kit was used (Figure 5C). Interestingly, the results indicated an upregulated mode of the AKT signaling pathway. DpC treatment led to an increase in activating phosphorylation of AKT 1/2/3 (S473) together with phosphorylation of its downstream effectors and an increase in AKT-mediated activating phosphorylation of PRAS40 (T246), inhibitory phosphorylation of GSK3α/β (S21/S9), and activating phosphorylation of WNK1 (T60). WNK1 activation is of particular interest as it is reported to play a role in oxidative stress response [43]. Similarly, upregulation of c-JUN phosphorylation has been noted as a response to ROS-generating, iron-chelation treatment [44], which suggests that different iron chelators indeed elicit a similar stress response.

In contrast to what was expected after observing EGFR downregulation, the activation of the AKT pathway in general led us to further investigate the manner of AKT activity in response to DpC treatment. Interestingly, a consistent pattern of regulation was detected across all neuroblastoma cell lines, with little variation among them. While the total AKT levels decreased by approximately 25% in response to DpC treatment (Figure 6A), the levels of phosphorylated AKT (S473) showed a marked increase (Figure 6B), consistent with the previous results from phospho-kinase array analysis. While the increase in AKT phosphorylation was significant only after the first 24 h of DpC application, the levels of pAKT remained elevated throughout the 48 h treatment in all studied cell lines, with the exception of the CHLA-15 cell line.

### 2.3. DpC Inhibits MYC Proteins Regardless of the MYCN Amplification Status

Although DpC-induced, HIF1-mediated expression of NDRG1 has already been elucidated, it appears that this is not the only mode of NDRG1 activation [45]. We report that DpC markedly decreases the levels of the well-established NDRG1 suppressor N-MYC [46] and its structural and functional homolog c-MYC [47,48]. Inhibition of N-MYC/c-MYC alleviated the suppression of the NDRG1 promoter and further enhanced the HIF1-mediated expression of NDRG1. Indeed, DpC treatment dramatically reduced the N-MYC oncoprotein *MYCN*-amplified SK-N-BE(2) cell line, as well as c-MYC in all other *MYCN*-nonamplified cell lines, SH-SY5Y, CHLA-15, and CHLA-20 (Figure 7A; Appendix A). In the SH-SY5Y cell line, the nuclear localization of c-MYC was markedly decreased after treatment, thus validating the results obtained from immunoblotting analysis (Figure 7B).

To elucidate whether DpC downregulates N-MYC at the transcriptional level, qRT–PCR was performed. Significant downregulation of *MYCN* transcripts (*p* < 0.05) was detected in DpC-treated SK-N-BE(2) cells (Figure 7C), which suggests that DpC might exert its activity against neuroblastoma cells via transcriptional inhibition of MYC oncoproteins, which are major drivers of aggressive neuroblastoma.

### 2.4. DpC Induces Morphological Changes and Lipid Accumulation in Neuroblastoma Cells

An intriguing morphological change was detected during routine microscopic analysis of DpC-treated neuroblastoma cells. The cytoplasm of these cells contained large spherical bodies (Figure 8A), which were identified as neutral lipid droplets and confirmed by bright-field imaging after staining with lipophilic Oil Red O and through fluorescence imaging after staining with LipidTOX™ Green (Figure 8B). Interestingly, the accumulation of neutral lipid droplets has already been reported by Zirath et al. [49] in SK-N-BE(2) neuroblastoma cells treated with N-MYC inhibitors. Here, we observed an identical phenotype in yet another neuroblastoma cell line: *MYCN*-nonamplified SH-SY5Y (Figure 8A). When quantified by flow cytometry after neutral lipid staining by LipidTOX™ Green, DpC treatment resulted in a significant (approximately 6-fold) increase in lipid accumulation (Figure 8C). This shows that various modes of MYC protein inhibition result in identical phenotypes and suggests a common role of MYC proteins in lipid metabolism/trafficking.

## 3. Discussion

Elucidating the role of thiosemicarbazone treatment in neuroblastoma cell lines at the molecular level proved to be a natural continuation of several previous observations: DpC succeeded in inhibiting growth as well as inducing cell death in neuroblastoma cells and increasing the levels of caspase 3 and 9 as well as phosphorylated JNK [22]. Moreover, thiosemicarbazone treatment greatly increases NDRG1 levels [50], a positive prognostic factor in neuroblastoma [23]. DpC sequesters cellular iron, which drives cells into a hypoxia-like state, and subsequently activates HIF1, as previously described [40], which was in concordance with our observation in the SH-SY5Y cell line (Appendix A). However, HIF1-mediated induction of NDRG1 expression (i.e., HIF1 binding to the HIF1-responsive element of the NDRG1 promoter) does not seem to be the only mode of NDRG1 upregulation by thiosemicarbazone treatment [45]. Iron-chelation treatment by DFO was reported to inhibit the expression of a well-known NDRG1 suppressor, N-MYC [51].

For the first time, we hereby report that DpC also markedly decreases the levels of N-MYC, as well as its structural and functional homolog c-MYC, both of which are known to bind to the core promoter of NDRG1 in an MIZ-1-dependent manner [25]. Our results indicate that DpC-mediated inhibition of MYC proteins alleviated the suppression of NDRG1 expression and further enhanced the expression of NDRG1. In the *MYCN*-amplified SK-N-BE(2) cell line, the level of the N-MYC oncoprotein decreased greatly, as did c-MYC in all other *MYCN*-nonamplified cell lines (SH-SY5Y, CHLA-15, CHLA-20). On a subcellular level, nuclear c-MYC localization decreased greatly, which further confirms the effect of DpC on this transcription factor. Generally, iron chelators are known for their ability to induce cell arrest and inhibit the function of iron-dependent enzymes, such as ribonucleotide reductase and DNA polymerase α [52,53]. However, the inhibition of these enzymes was reportedly not responsible for the decrease in N-MYC levels in response to iron-chelation treatment [51]. The decrease in MYC proteins by DpC thus seems to be facilitated through a different mechanism.

Downregulation of MYC proteins bears significance beyond facilitating the liberation of NDRG1 expression. Deregulated in over half of all human cancers [54] and established as one of the prominent cancer stem cell markers in neuroblastoma [55], MYC oncoproteins represent an intriguing target for cancer therapy. DpC treatment significantly decreased MYC protein levels across all neuroblastoma cell lines, which suggests promising new therapy strategies, especially in *MYCN*-amplified neuroblastomas, which account for approximately 20% of cases and rank among those with the most difficult course of the disease [1]. DpC suppressed *MYCN* expression and total protein levels, which proves its multimodal effect on one of the most prominent prognostic factors in neuroblastoma.

DpC succeeded in downregulating the levels of EGFR in all four studied neuroblastoma cell lines; thus we identified another shared effect of thiosemicarbazone treatment in this type of cancer. Moreover, in the SH-SY5Y neuroblastoma cell line, DpC was found to inhibit EGFR through NDRG1 in accordance with Menezes et al. [35], who partially elucidated the role of NDRG1 in EGFR regulation in pancreatic carcinoma cell lines. A decrease in the phosphorylated forms of EGFR (pY845, pY992, pY1045, pY1068, and pY1148) suggests inhibition of downstream signal transduction at multiple levels. However, we were unable to detect MIG-6, which has been reported to be responsible for NDRG1-mediated EGFR downregulation in pancreatic carcinoma cell lines (data not shown), which illustrates the need for further study of the means of action. Additionally, in the SK-N-BE(2) neuroblastoma cell line, silencing NDRG1 did not rescue EGFR downregulation, thus suggesting a different mechanism than via NDRG1.

Following downstream of EGFR, we explored two of the most important EGFR-mediated signaling pathways: the PI3K–AKT and the RAF–MEK–ERK axes. While the overall decrease in total AKT protein levels across all studied cell lines would suggest the downregulation of this pathway, the levels of its phosphorylated/active form increased in all reference cell lines. The upregulation of the pro-survival AKT pathway is indeed in stark contrast to what would be expected, considering the cytotoxic effect that the DpC treatment exerted on neuroblastoma cells. Two possible explanations emerge to clarify such observation: (1) the cytotoxic effects of DpC could be responsible for cell death in spite of the up-regulated pro-survival AKT pathway. In this scenario, pAKT upregulation would work against the intended effect of DpC and thus elevated pAKT signaling would actually confer resistance to the genotoxic treatment [56]. However, a study by Lui et al. [57] elucidated that while there was an increase in pAKT following thiosemicarbazone treatment, the activation of its downstream effectors (mTOR, Cyclin D) expression was not observed. (2) Overly up-regulated AKT signaling could induce cell death by sensitizing neuroblastoma cells to ROS, which are generated in the process of DpC treatment. While some reports can be found to support this claim [58,59], the elucidation of AKT dynamic in response to treatment by thiosemicarbazones in neuroblastoma requires further research. As for ERK-bound signalization, the unhindered levels of total and phosphorylated ERK (Appendix A) strongly suggest its lateral activation, which compensates for the loss of EGFR signaling [60].

The induction of lipid accumulation in response to chelation therapy proved to be an unexpected discovery that warrants further research for multiple reasons.

First, it coincides with the identical phenotype that was observed by Zirath et al. [49], who used a specific N-MYC inhibitor in the SK-N-BE(2) neuroblastoma cell line. We expanded this by observing lipid accumulation in yet another neuroblastoma cell line, SH-SY5Y. Even though this cell line is *MYCN*-nonamplified, its structural and functional homolog, c-MYC, is downregulated in response to DpC treatment. Therefore, direct inhibition of c-MYC in SH-SY5Y cells would not only further confirm the functional overlap in MYC proteins but also further establish their role in lipid accumulation in neuroblastoma.

Second, since MYC proteins are direct transcriptional inhibitors of NDRG1, any inhibition of MYC protein expression (whether through DpC application, knockout, or direct small-molecule inhibitors) likely results in NDRG1 expression liberation and upregulation. Stacking evidence across cell/tissue types indicates that NDRG1 could play a major role in facilitating lipid accumulation in the following ways. (i) In oligodendrocytes, NDRG1 confers LDL uptake by regulating endosomal trafficking of LDL-R. Silencing NDRG1 indeed resulted in a reduction in lipid uptake by cells [61]. (ii) In Schwann cells, NDRG1 mutation manifests as a rare demyelinating neuropathy (Charcot–Marie–Tooth disease type 4D) characteristic of progressive hypomyelination and axonal loss [62]. Schwann cells share a neural crest origin with neuroblastoma cells, thus indicating that NDRG1 could play a similar role in lipid metabolism in these cell types. (iii) In adipocytes, silencing NDRG1 results in decreased lipidogenesis in differentiated cells [63]. While these findings indicate that NDRG1 plays a pivotal role in cellular lipid uptake in neuroblastoma, further investigation is needed to fully explain the exact mechanism of NDRG1 action.

Overall, iron chelators treatment by the novel thiosemicarbazone DpC induces potent cell death of neuroblastoma cells, while targeting some of the major drivers of aggressive neuroblastomas, regardless of their *MYCN* status (Figure 9). DpC could thus be a useful addition in the treatment of MYC-driven neuroblastomas.

## 4. Materials and Methods

### 4.1. Cell Culture

Four neuroblastoma cell lines used in this study were selected based on their genetic background or treatment status to better cover the heterogeneity observed among neuroblastomas. The *MYCN*-nonamplified SH-SY5Y (No. 94030304) and *MYCN*-amplified SK-N-BE(2) (No. 95011815) cell lines were obtained from ECACC (Salisbury, UK). To analyze postrelapse changes in neuroblastoma pathophysiology, the CHLA-15 cell line was derived from initial surgical biopsy and the CHLA-20 cell line was derived from the posttreatment relapse resection of neuroblastoma in the same patient. Both of these cell lines were obtained from Alex’s Lemonade Stand Foundation Childhood Cancer Repository (cccells.org) and kindly provided by Dr. Michael D. Hogarty (Children’s Hospital of Philadelphia, Philadelphia, PA, USA).

Cells were grown in Dulbecco’s modified Eagle’s medium-F12 medium mixture (DMEM-F12, GE Healthcare, Chicago, IL, USA) supplemented with 10% (CHLA 15, CHLA 20) or 20% (SH-SY5Y, SK-N-BE(2)) fetal bovine serum (FBS), 100 IU/mL penicillin, 100 IU/mL streptomycin, 2 mM glutamine, and 1% MEM amino acid solution (all from Biosera, Nuaillé, France). The medium for the CHLA-15 and CHLA-20 cell lines was further supplemented with 1× ITSX (insulin–transferrin–selenium) solution (Thermo Fisher Scientific, Waltham, MA, USA). Cell culture was performed under standard conditions at 37 °C and 95% humidity in 5% CO_2_.

### 4.2. DpC Treatment

Di-2-pyridylketone 4-cyclohexyl-4-methyl-3-thiosemicarbazone (DpC) was obtained from Sigma–Aldrich (St. Louis, MO, USA, Cat. No. SML0483), diluted with DMSO (Sigma–Aldrich) to a 100 mM stock solution, aliquoted, and stored at 20 °C. For all experiments, cells were treated with DpC 24 h after being seeded onto Petri dishes so that the confluence reached 70%. Diluting DpC from the 100 mM stock solution resulted in a final DpC concentration of 20 μM (SH-SY5Y, SK-N-BE(2)) or 2 μM (CHLA 15, CHLA 20), while the concentration of DMSO did not exceed 0.02% or 0.002%, respectively.

### 4.3. Western Blot Analysis

Protein extracts were isolated from cell lysates and stored at −80 °C. Prior to loading, the protein concentration was determined using the DC™ Protein Assay Kit (Bio–Rad Laboratories, Hercules, CA, USA) as per the manufacturer’s protocol. Protein samples were loaded into 10% polyacrylamide gels, electrophoresed, and blotted onto PVDF membranes. Following membrane blocking (with either 5% bovine serum albumin or 5% nonfat dry milk solution in a TBS–Tween buffer), the membranes were incubated with primary antibodies overnight at 4 °C (Appendix A). Mouse or rabbit IgG horseradish peroxidase-conjugated antibodies served as secondary antibodies (Appendix A). Chemiluminescence was induced by an ECL-Plus detection kit (GE Healthcare), detected by an Azure c600 imaging system (Azure Biosystems, Dublin, CA, USA) or AGFA CP-BU X-ray films (AGFA, Mortsel, Belgium) and quantified using ImageJ software v1.52a (U. S. National Institutes of Health, Bethesda, MD, USA) [64]. All experiments were performed in biological triplicate.

### 4.4. siRNA Silencing Assay

Cells were cultivated in 60 mm Petri dishes in a complete DMEM-F12 medium and transfected using a Lipofectamine^®^ RNAiMAX and Silencer™ Pre-Designed siRNA mixture in Gibco™ Opti-MEM™ Medium (Thermo Fisher Scientific) according to the manufacturer’s instructions. Two distinct siRNAs (Thermo Fisher Scientific; siRNA1 ID: 135611; siRNA2 ID: 135612) and one scramble negative control siRNA (Thermo Fisher Scientific; ID: 4390843) were used in the assay. Cells were harvested 72 h after transfection, followed by lysate preparation and protein extraction.

### 4.5. Immunofluorescence Staining

To perform the immunofluorescence assay, cells were seeded onto coverslips in 35 mm Petri dishes and cultured for 24 h so that the confluence reached 70%. At this confluence, treatment was added to the medium for 48 h. The cells were then washed with PBS, fixed with 3% paraformaldehyde (Sigma–Aldrich), and permeabilized with 0.2% Triton X-100 (Sigma–Aldrich). Afterward, the cells were rinsed with PBS and blocked with 3% bovine serum albumin solution in PBS. The cells were incubated with either primary or fluorophore-conjugated antibodies in a humidified chamber for 60 min at 37 °C (Appendix A). After the cells were thoroughly rinsed with PBS, secondary antibodies were added to the corresponding primary antibody for another 45 min at 37 °C in a humidified chamber (Appendix A). Cell nuclei were counterstained with 0.05% Hoechst 33342 (Life Technologies, Carlsbad, CA, USA). The coverslips were mounted onto microscope slides with ProLong™ Diamond Antifade Mountant (Thermo Fisher Scientific). Negative controls were prepared by either using a fluorophore-conjugated isotype control antibody or omitting the primary antibodies. An Olympus BX-51 microscope with an Olympus DP72 CCD camera was used to capture micrographs, which were then analyzed by Cell^P v2.3 imaging software (Olympus, Tokyo, Japan). Additionally, confocal microscopy was employed to acquire high-resolution micrographs using a Leica SP8 confocal microscope (Leica Microsystems, Wetzlar, Germany). LAS X v5.0.3 imaging software (Leica Microsystems) was used for the subsequent graphic analysis of micrographs.

### 4.6. qRT–PCR

Total RNA was extracted and transcribed into cDNA as described by Skoda et al. [65]. The expression of *MYCN* (F: 5′-AGAGGACACCCTGAGCGATT-3′; R: 5′-GGTGAATGTGGTGACAGCCT-3′) in response to DpC treatment was performed in technical and biological triplicates (i.e., different cell passages in vitro). The heat shock protein gene HSP90AB1 (F: 5′-CGCATGAAGGAGACACAGAA-3′; R: 5′-TCCCATCAAATTCCTTGAGC-3′) was used as the endogenous reference control.

### 4.7. Neutral Lipid Staining

For bright-field microscopy analysis, cells were fixed with 3% paraformaldehyde, rinsed with PBS, stained with lipophilic Oil Red O for 20 min, washed with PBS, and mounted onto slides. For immunofluorescence analysis, cells were fixed with 3% paraformaldehyde, rinsed with PBS, and treated with a 1:500 solution of LipidTOX™ Green for 30 min at room temperature, followed by another wash with PBS. For flow cytometry analysis, the cell suspension was fixed with 3% paraformaldehyde, washed with PBS, and treated with a 1:1000 solution of LipidTOX™ Green for 30 min, followed by another wash with PBS.

### 4.8. Immunostaining for Flow Cytometry

Cell suspensions were fixed with 3% paraformaldehyde for 20 min, permeabilized with 0.1% Triton X-100 (Sigma–Aldrich) for 1 min, blocked with 3% bovine serum albumin, and incubated with fluorophore-conjugated antibodies or their respective isotype control for 1 h at 37 °C (Appendix A). The cell suspension was rinsed with PBS between each step. Cytometric analysis was performed using CytoFLEX S (Beckmann Coulter, Brea, CA, USA), and 1 × 104 valid events were evaluated by CytExpert v2.4 software (Beckmann Coulter).

### 4.9. Phospho-Protein and Cell Stress Protein Array Analysis

Relative total and/or phosphorylation levels of selected target proteins involved in signal transduction and the cell stress response were analyzed using the following protein array kits: Proteome Profiler™ Human Cell Stress Array Kit (Appendix A) and Proteome Profiler™ Human Phospho-Kinase Array Kit (Appendix A) (both R&D Systems, Minneapolis, MN, USA). The samples were processed according to the manufacturer’s protocol, and the chemiluminescence signal was quantified using ImageJ software v1.52a (U. S. National Institutes of Health) [64] and analyzed in concordance with our previous study [66].

### 4.10. Statistical Analysis

Relative density analyses of immunoblotting were normalized with regard to their respective GAPDH loading control. Relative pixel density (Western blot), fold change (qRT-PCR), as well as median fluorescence intensity (flow cytometry) were evaluated using two-tailed unpaired *t*-test: * *p* < 0.05 and ** *p* < 0.01 were considered statistically significant.

## 5. Conclusions

This study built upon the previous findings of thiosemicarbazone treatment in neuroblastoma and elucidated its effect on some of the important signaling molecules in this cancer type, namely MYC proteins, EGFR and its downstream targets, or NDRG1, whose exact mechanism of action remains largely elusive. As it entered Phase I clinical trials, DpC held promise to become a new addition to certain oncotherapy protocols. Despite DpC exhibiting superior pharmacokinetic properties (e.g., increased half-life and resistance against inactivation by N-demethylation) over its predecessor Dp44mT [67], clinical trials have reported myalgia in patients treated with DpC [68]. Nevertheless, iron chelators still show promising results in potentially treating certain types of neuroblastoma, particularly those with upregulated expression of N-MYC.

Specifically, treatment with DpC proved to target some of the major molecular targets in neuroblastoma, namely the MYC proteins and both total and phosphorylated levels of EGFR. Regulation of EGFR is at least partially mediated by NDRG1, which is in accordance with the results observed in pancreatic carcinoma. Furthermore, the formation of lipid droplets after DpC treatment (and subsequent MYC downregulation) in the SH-SY5Y cell line expanded upon the induction of an identical phenotype in the SK-N-BE(2) cell line in response to N-MYC inhibition.

## Figures and Tables

**Figure 1 ijms-23-00376-f001:**
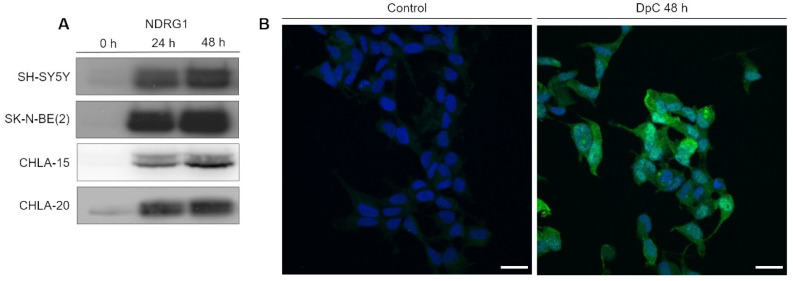
NDRG1 is upregulated in neuroblastoma cells in response to iron-chelation therapy by DpC. (**A**) Immunoblotting of NDRG1 protein levels in neuroblastoma cell lines treated with 20 μM (SH-SY5Y, SK-N-BE(2)) or 2 μM (CHLA-15, CHLA-20) DpC. Representative images of three independent experiments. Source data are provided in Appendix A. (**B**) Massive increase of NDRG1 (green) expression in DpC-treated cells in comparison to control. Immunofluorescence micrographs of SH-SY5Y cells treated with 20 μM DpC for 48 h. Nuclei counterstained with Hoechst 33342 (blue). Scale bar: 20 μm.

**Figure 2 ijms-23-00376-f002:**
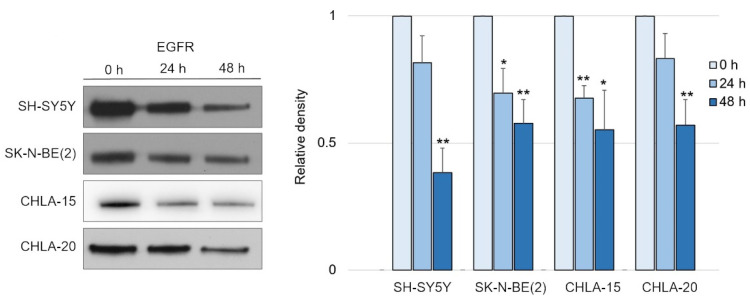
EGFR is downregulated in response to iron-chelation therapy by DpC. (**A**) Immunoblotting of EGFR protein levels in neuroblastoma cell lines treated with 20 μM (SH-SY5Y, SK-N-BE(2)) or 2 μM (CHLA-15, CHLA-20) DpC. Representative images (**left**) and relative optical density values (**right**) of three independent experiments. Source data are provided in Appendix A. Densitometry data are shown as the mean ± SD normalized to 0 h values. * *p* < 0.05; ** *p* < 0.01, two-tailed unpaired *t*-test.

**Figure 3 ijms-23-00376-f003:**
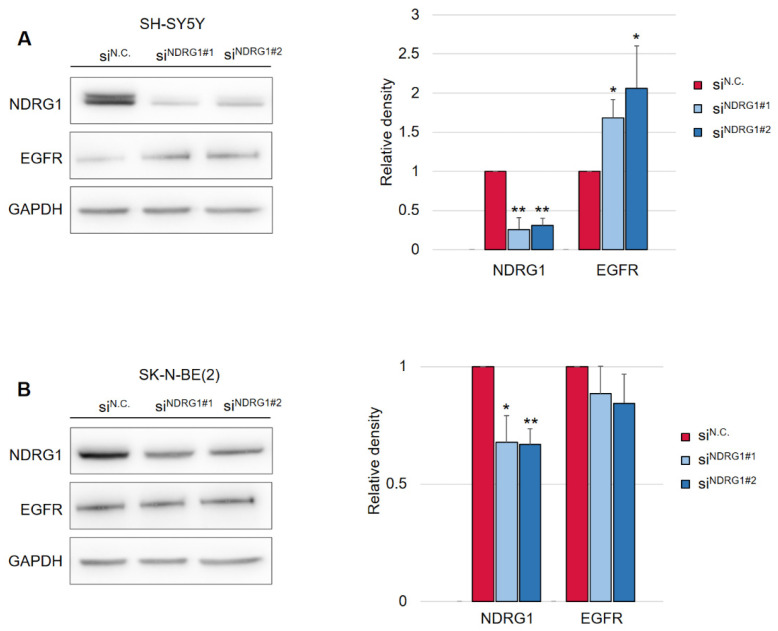
Silencing of NDRG1 rescues the effect of DpC on total levels of EGFR in the SH-SY5Y cell line but not in SK-N-BE(2). (**A**) Immunoblotting of NDRG1 and EGFR protein levels in SH-SY5Y cells treated with 20 μM DpC for 48 h and specific siRNAs for NDRG1 (siNDRG1#1 and siNDRG1#2) or negative silencer control (siN.C.). (**B**) Immunoblotting of NDRG1 and EGFR protein levels in SK-N-BE(2) cells treated with 20 μM DpC for 48 h and specific siRNAs for NDRG1 (siNDRG1#1 and siNDRG1#2) or negative silencer control (siN.C.). Representative images (**left**) and relative optical density values (**right**) of three independent experiments. Densitometry data are shown as the mean ± SD normalized to siN.C. values. * *p* < 0.05; ** *p* < 0.01; two-tailed unpaired *t*-test.

**Figure 4 ijms-23-00376-f004:**
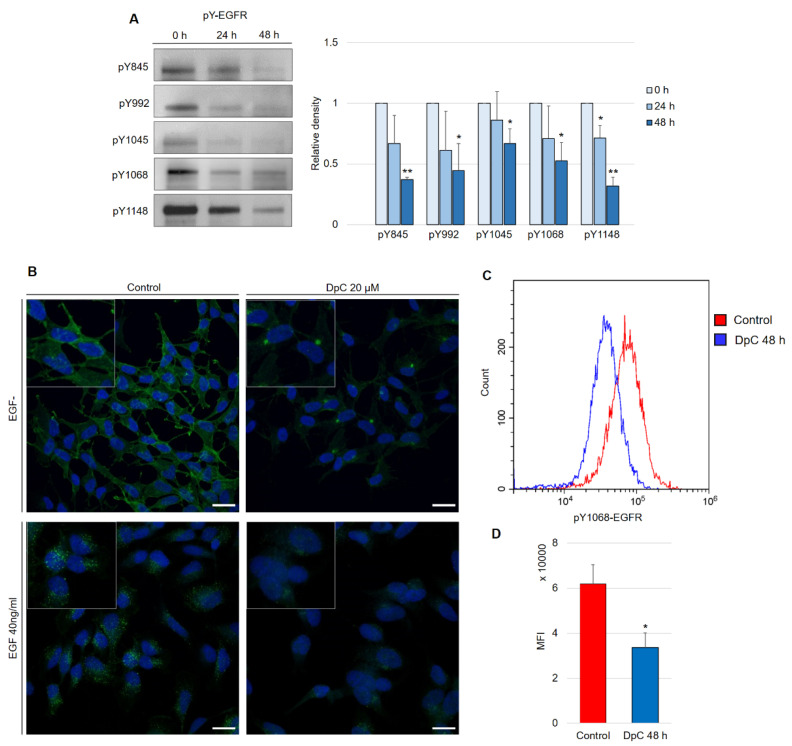
DpC downregulates the levels of phosphotyrosine-EGFR. (**A**) Immunoblotting of phosphorylated forms of EGFR in the SH-SY5Y neuroblastoma cell line treated with 20 μM DpC. Representative images (**left**) and relative optical density values (**right**) of three independent experiments. Source data are provided in Appendix A. Densitometry data are shown as the mean ± SD normalized to 0 h values. * *p* < 0.05; ** *p* < 0.01; two-tailed unpaired *t*-test. (**B**) Immunofluorescence micrographs of EGFR (green) in SH-SY5Y cells treated with 20 μM DpC and EGF (40 ng/mL; 10 min; 37 °C). Nuclei counterstained with Hoechst 33342 (blue). Scale bar: 20 μm. (**C**,**D**) Quantification of pY1068-EGFR protein levels in response to DpC treatment (20 μM; 48 h) in SH-SY5Y cells. Median fluorescence intensity (MFI) ± SD comprised of three independent experiments. * *p* < 0.05; ** *p* < 0.01; two-tailed unpaired *t*-test.

**Figure 5 ijms-23-00376-f005:**
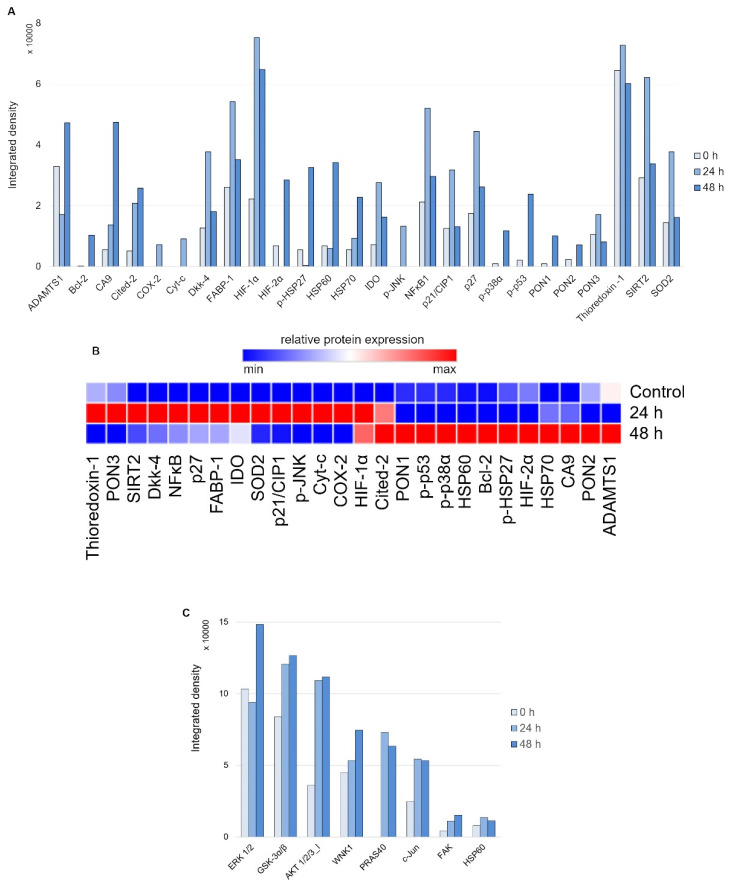
Array profiling of stress response proteins and kinase phosphorylation in response to DpC in the SH-SY5Y cell line. (**A**) Stress response protein profile of cells treated with 20 μM DpC for 24 and 48 h. (**B**) Hierarchical clustering of relative stress protein expression. (**C**) Phospho-kinase protein profile of cells treated with 20 μM DpC for 24 and 48 h.

**Figure 6 ijms-23-00376-f006:**
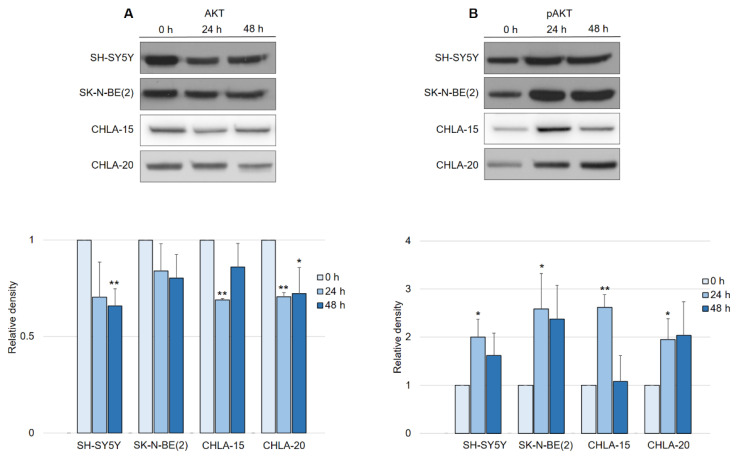
DpC regulates total and phosphorylated AKT protein levels in neuroblastoma cell lines. (**A**) Immunoblotting of AKT and (**B**) immunoblotting of pAKT (S473) protein levels in neuroblastoma cells treated with 20 μM (SH-SY5Y, SK-N-BE(2)) or 2 μM (CHLA-15, CHLA-20) DpC. Representative images (**up**) and relative optical density values (**down**) of three independent experiments. Source data are provided in Appendix A. Densitometry data are shown as the mean ± SD normalized to 0 h values. * *p* < 0.05; ** *p* < 0.01; two-tailed unpaired *t*-test.

**Figure 7 ijms-23-00376-f007:**
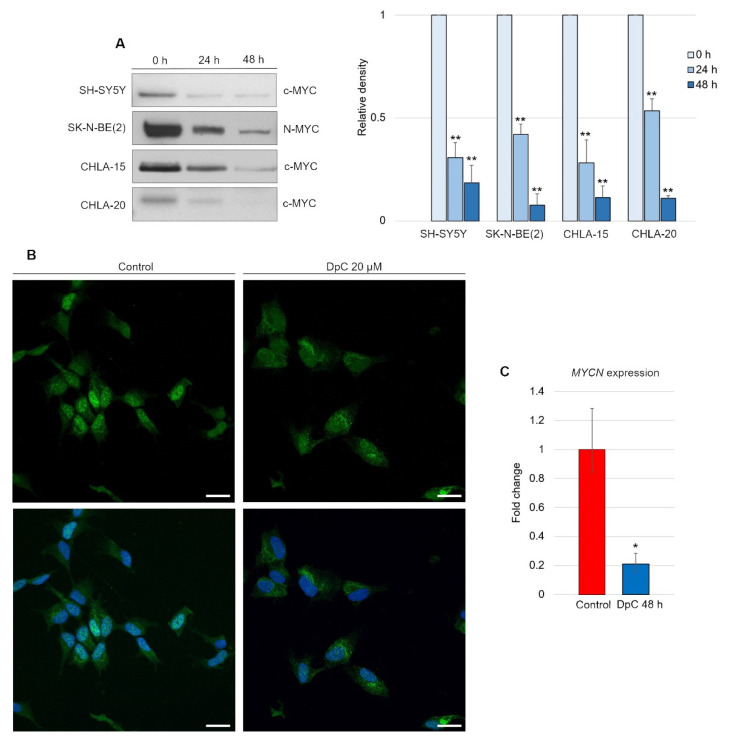
DpC downregulates MYC proteins in neuroblastoma cell lines. (**A**) Immunoblotting of c-MYC (SH-SY5Y, CHLA-15, CHLA-20) and N-MYC (SK-N-BE(2)) protein levels in neuroblastoma cells treated with 20 μM (SH-SY5Y) or 2 μM (CHLA-15, CHLA-20) DpC. Representative images (**left**) and relative optical density values (**right**) of three independent experiments. Source data are provided in Appendix A. Densitometry data are shown as the mean ± SD normalized to 0 h values. * *p* < 0.05; ** *p* < 0.01; two-tailed unpaired *t*-test. (**B**) Immunofluorescence micrographs of c-MYC (green) in SH-SY5Y cells treated with 20 μM DpC and EGF (40 ng/mL; 10 min; 37 °C). Nuclei counterstained with Hoechst 33342 (blue). Scale bar = 20 μm. (**C**) qRT–PCR analysis of the effect of 20 μM DpC on the expression of *MYCN* in the SK-N-BE(2) cell line. Fold change ± SD comprised of three independent experiments. * *p* < 0.05; ** *p* < 0.01; two-tailed unpaired *t*-test.

**Figure 8 ijms-23-00376-f008:**
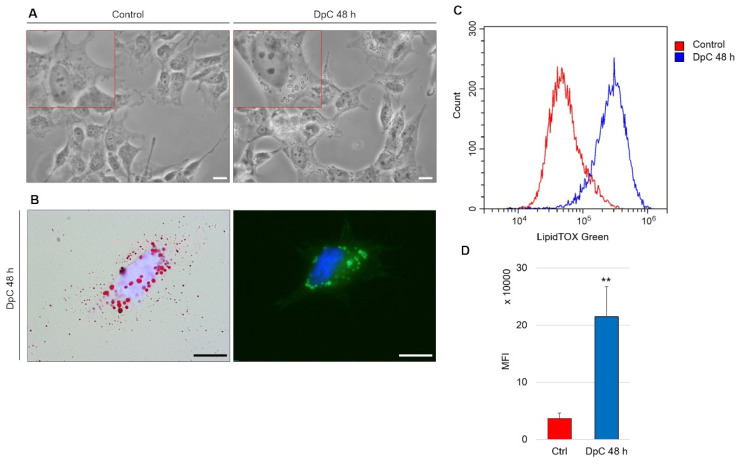
Lipid accumulation in SH-SY5Y neuroblastoma cells in response to DpC treatment. (**A**) Light-microscopy micrograms revealing spherical bodies in the cytoplasm of SH-SY5Y cells. (**B**) Neutral lipid staining of DpC-treated cells by Oil Red O (red, **left**) and LipidTOX™ Green (green, **right**). Nuclei counterstained with Hoechst 33342 (blue). Scale bar: 20 μm. (**C**,**D**) Increase in the median fluorescence intensity (MFI) of LipidTOX™ Green emission after DpC treatment. MFI ± SD comprised of three independent experiments. ** *p* < 0.01.

**Figure 9 ijms-23-00376-f009:**
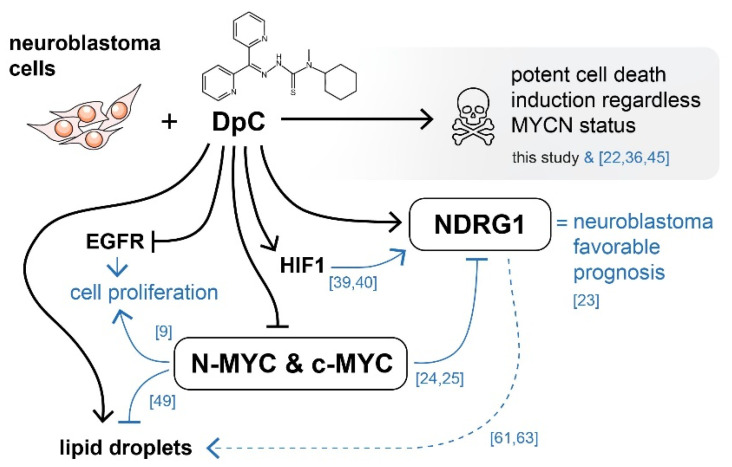
Graphical overview of the major molecular targets affected by DpC treatment in neuroblastoma. Black arrows indicate findings of this study. Blue arrows represent existing knowledge (see references), providing a mechanistic insight into the effects of thiosemicarbazones in the context of neuroblastoma cells. DpC suppressed the expression of EGFR and MYC proteins in all studied neuroblastoma cell lines. Additionally, the treatment induced a massive increase in NDRG1 levels, as well as lipid droplet accumulation. Although the involvement of NDRG1 in lipid accumulation has been suggested in neuroblastoma [61] and other cell types [62,63], the underlying mechanism is still unclear (dashed line).

## Data Availability

The data presented in this study are available in the article and the Appendix A.

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
