# Peer review of "Iron-Chelation Treatment by Novel Thiosemicarbazone Targets Major Signaling Pathways in Neuroblastoma"

_ijms, 2021, doi:10.3390/ijms23010376_

Round 1
Reviewer 1 Report
The authors show that treatment of neuroblastoma cells with DpC suppresses the expression of MYC family protein. This is a very interesting observation and I appreciate the analysis of the molecular mechanism. However, some of the following points should be clarified before the publication.
Major comments
- The effects of DpC on neuroblastoma cell proliferation, migration, invasion, and/or cell death have not been evaluated. What cell phenotypes does DpC affect, and how are these phenotypic effects explained by the molecular mechanisms identified by the authors?
- In Figure 3B, knockdown of NDRG1 is less efficient than in Figure 3A, and this technical problem may be the reason why EGFR is not significantly increased. The authors should examine this possibility. Previous reports have also shown that MYCN and EGFR are mutually regulated in MYCN-amplified neuroblastoma (Hossain et al. Cancer Res. 2012 Sep 1;72(17):4587-96; Pan Front Oncol. 2021 Apr 21;11:633579), and examination of MYCN and cMYC expression levels may explain this cell type-specific result.
- Is the relative density shown in the western blotting results a value that represents a ratio to the loading control? In particular, in Figure 4A, the density of phosphorylated EGFR divided by the density of total EGFR should be used to evaluate the phosphorylation levels.
- It has been reported that GSK3beta promotes T58 phosphorylation of MYCN and contributes to the degradation of MYC and MYCN. It has also been reported that phosphorylated AKT promotes S9 phosphorylation of GSK3beta and contributes to the stabilization of MYC and MYCN. These reports indicate that the results shown in Figures 5 and 6 and the decrease in MYCN and MYC protein levels shown in Figure 7A are seemingly contradictory. The authors should show in Figure 7A whether phosphorylation of S9 of GSK3beta occurs with GSK3beta-specific antibodies. In addition, MYCN itself is reported to induce MYCN Therefore, the transcriptional repression of MYCN may be a secondary effect caused by MYCN degradation. It should be tested whether the reduction of MYC and MYCN proteins can be inhibited by proteasome inhibitors.
Minor comments
The legend of Figure 7A is wrong and it seems to be MYC or MYCN, not AKT. Is the SH-SY5Y data in Figure 7A a quantification of MYCN or MYC expression?
Reviewer 2 Report
It is an interesting article by Macsek et al. They reported the effect of Iron-chelation treatment (DPC) on various neuroblastoma cell lines. DPC suppresses many tumor-promoting proteins and upregulated proteins that have anti-tumor effects. These findings might have clinical implications.
However, there is significant concern about the data presented, and experiments have been done
Please define the functions MYCN, N-MYC, and c-MYC in neuroblastoma in the introduction section.
Why did the authors use two different concentrations of DPC (20 μM & 2 μM) for treating the different cell lines?
This reviewer did not understand how authors calculate the SD/SEM with a single sample; indeed, the reviewer did not mention it in the statistical analysis section.
Did the authors repeat any of the experiments?
Is there any effect of this treatment on primary human cells?
What is the significance of lipid accumulation in neuroblastoma cells?
Figure 7 B is not labeled correctly
Did the authors find any anti-proliferative or apoptotic effect on neuroblastoma cell lines in response to DPC?
Please discuss, why there is upregulation of p-AKT but not total AKT?
This paper should have graphical abstract to understand the mechanism.
Round 2
Reviewer 1 Report
The authors have responded accurately to my comments, and I believe their paper has improved enough to be published.
Author Response
We are very thankful for your positive comment. We sincerely hope this article will be highly accessed and helpful for readers.
Reviewer 2 Report
Please explain xx, yy in figure 9.
